# Coupled Field Analysis of Phenomena in Hybrid Excited Magnetorheological Fluid Brake

**DOI:** 10.3390/s23010358

**Published:** 2022-12-29

**Authors:** Wojciech Szelag, Cezary Jedryczka, Adam Myszkowski, Rafal M. Wojciechowski

**Affiliations:** 1Institute of Electrical Engineering and Electronics, Poznan University of Technology, 60-965 Poznan, Poland; 2Institute of Mechanical Technology, Poznan University of Technology, 60-965 Poznan, Poland

**Keywords:** brake, magnetorheological fluids, coupled phenomena, electromagnetic fields, fluid dynamics, thermal field

## Abstract

The paper presents a field model of coupled phenomena occurring in an axisymmetric magnetorheological brake. The coupling between transient fluid dynamics and electromagnetic and thermal fields as well as mechanical equilibrium equations is taken into account. The magnetic field in the studied brake is of an excited hybrid manner, i.e., by the permanent magnets (PMs) and current *I_s_* in the excitation winding. The finite element method and a step-by-step algorithm have been implemented in the proposed field model of coupled phenomena in the considered brake. The nonlinearity of the magnetic circuit and rheological properties of a magnetorheological fluid (MR fluid) as well as the influence of temperature on the properties of materials have been taken into account. To solve equations of the obtained field model, the Newton–Raphson method and the coupled block over-relaxation method have been implemented. The elaborated algorithm has been successfully used in the analysis of the phenomena in the considered magnetorheological brake. The accuracy of the developed model and its usefulness have been verified by a comparative analysis of the results of simulation and laboratory tests carried out for the developed prototype of the studied brake.

## 1. Introduction

The paper presents the authors’ concept of a hybrid excited magnetorheological brake (HEMRFB) (Figure 1), developed to work in modern elevator drive systems [1]. The operating principle of the proposed transducer is based on the phenomenon of viscosity and the yield stress change of the MR fluid subjected to a magnetic field [1,2,3,4]. The first MR fluids were developed in the 1930s by Jacob Rabinow [5]. The change in viscosity in the effect of the action (under the influence) of the magnetic field is a characteristic feature of these magnetically controllable fluids. A change in fluid viscosity is inseparably connected with a change in yield stress τ_0_ between infinite layers of the fluid [6,7,8]. Owing to their properties, MR fluids, in recent years, have been applied for their efficient control of torque and force transmission in new types of so-called magnetorheological transducers. They are very often used, among others, in the development of new types of brakes and dampers, as well as clutches [4,9,10,11,12,13,14,15,16,17], offering an alternative way for controlling torque and force transmission in well-known electromechanical transducers. The magnetic field in the area with an MR fluid in the hybrid excited brake presented here is produced by both the arrangement of two windings and a system of permanent magnets (Figure 1). This field in the magnetic circuit of the brake is controlled by the changes in the value currents in both excitation windings. The increase in the braking torque of the tested brake is due to the rise in fluid viscosity caused by the increase in magnetic field density in the MR fluid region and allows for precise control of the braking torque by the brake winding supply current. The unique feature of the proposed hybrid excited brake is the generation of antitorque without current in the excitation winding [1,18,19]. To disengage the brake, i.e., while the elevator car is moving, the windings are energized, and their DC currents are of such value and direction as to suppress the magnetic field in the gap with an MR fluid by redirecting the magnetic flux to the alternative path through the auxiliary air gaps. Under these conditions, the brake torque is very low. When the car reaches the desired stop, the excitation windings are turned off, resulting in a high counter torque in the brake. The main purpose of the brake is to prevent any movement of the elevator car during a standstill. However, the proposed HEMRFB can also be used as an emergency brake. As the antitorque of the brake can be controlled by changing the value of the currents in the excitation windings, the design of the magnetic circuit allows one to obtain a greater antitorque value than the antitorque corresponding to the current *I* = 0 A. This operation range can be used during emergency braking. The proposed HEMRFB has many advantages such as quiet operation, lower power consumption than in the electromagnet-driven brakes, easy braking torque regulation, and an ability to achieve quasilinear control characteristics by proper design of the magnetic circuit [19].

Electromagnetic devices with an MR fluid are investigated in many scientific institutions around the world. The research work conducted at these institutions focuses on the analysis of the operating states of existing devices and on methods to improve their functional performance. At the same time, work is also underway to develop new designs of devices [9,12,13,20,21,22,23]. In order to increase the accuracy of the obtained results of the analysis carried out for the operation states of electromechanical devices with an MR fluid, the rheological properties of the fluid, winding resistance, resistivity of massive conductive elements, and magnetic properties of materials must be considered as temperature dependent [24,25,26,27,28]. This issue is very important when designing devices with an MR fluid, such as brakes, clutches, or dampers [20,29,30]. During operation, these devices can easily exceed the allowable temperature of the fluid and destroy its structure [31,32,33].

In the paper, a field model of coupled hydrodynamic, electromagnetic, thermal, and mechanical phenomena has been proposed and discussed. The proposed model has been developed to analyze the steady and transient states of the axisymmetric MR brake with a hybrid excitation system. In order to verify the proposed field model of coupled phenomena in the considered device, an elaborated algorithm and software, as well as a prototype of the brake, have been designed and tested.

## 2. Coupled Field Model of Phenomena in HEMRFB

Because of the high nonlinearities of the magnetic circuit and rheological properties of MR fluids, the phenomena occurring in devices with an MR fluid should be considered in terms of fields. In the MR brake considered, the velocity field of the fluid is dependent on the angular rotor velocity, as well as the yield stress distribution in the MR fluid. This shear stress in the fluid is dependent on both the distribution of magnetic flux density in the area with the fluid due to its rheological properties and the field of deformation of the fluid’s velocity. The resulting shear stress counteracts the rotor motion in the MR brake and is directly related to brake antitorque. Therefore, in the numerical model of an MR transducer, the stresses in the fluid and its velocity field should be coupled with the electromagnetic field. It should be emphasized that this coupling is bidirectional since the value of the transmitted torque influences the dynamics of the moving parts of the brake. Moreover, it should be highlighted that the fluid dynamic and electromagnetic phenomena in MR transducers are highly affected by the dependence of material properties on temperature. In order to precisely reproduce the phenomena, the conductivity of the regions with eddy currents, their magnetic permeability, winding resistance, and magnetic flux produced by the permanent magnets, as well as the viscosity and yield stress of an MR fluid, should be considered as a function of the temperature distribution.

Due to the strong coupling between the electromagnetic thermal and fluid velocity fields, the analysis of the phenomena in devices with MR fluids is a challenging and complex task in terms of the numerical implementation of the proposed models. Even more complicated is the transient nature of these fields and the nonlinearity of the equations describing them. So far, a comprehensive approach to the problem of solving time-dependent coupled field phenomena in MR devices has not yet been the subject of research discussed in the open literature. It can be observed that in the research area subjected to the analysis of MR transducers, the field approach is applied only for selected phenomena (for example, electromagnetic), while simplified models of fluid dynamics or temperature fields are incorporated [9,17,20,24,34].

Only one paper [33] presents a model that captures all electromagnetic, thermal, and fluid flow phenomena in a field manner. Unfortunately, this model is not adapted to the analysis of devices consisting of permanent magnets such as MR brakes with hybrid excitation. Therefore, in this paper, the authors have attempted to develop a complex coupled field model of phenomena in the considered MR brake with a hybrid excitation circuit, i.e., a coupled field model that takes into account the electromagnetic, ferrohydrodynamic, and thermodynamic phenomena together with the dynamics of movable parts of the studied MR brake.

In this paper, an MR brake with an axisymmetric structure has been considered (see Figure 1). Therefore, for the analysis of the considered brake with an MR fluid, a cylindrical coordinate system *r*, *z*, υ has been introduced—see Figure 2.

The equation system describing the transient electromagnetic field can be given in the following form [3,35,36,37]:(1)$$\frac{\partial }{\partial r}\left(\frac{1}{\mu l}\frac{\partial \varphi }{\partial r}\right)+\frac{\partial }{\partial z}\left(\frac{1}{\mu l}\frac{\partial \varphi }{\partial z}\right)=-J_{w}+\frac{\partial M_{r}}{\partial z}-\frac{\partial M_{z}}{\partial r}+\frac{\gamma }{l}\frac{d\varphi }{dt},$$
where *l* = 2π*r*, φ = 2π*rA*_υ_, *A*_υ_ is the component of magnetic vector potential **A** in the υ-direction, *J_w_ = i/s_c_* is the current density in the excitation winding, *i* is the current value in the winding, *s_c_* is the cross-section area of the winding wire, *M_r_* and *M_z_* are the components of magnetization vector **M** in the regions with PMs, μ is the magnetic permeability of the medium, and γ is the conductivity of the region with the eddy currents. For the MR fluid, the conductivity γ is equal to 0.

The magnetization characteristics of the considered fluids are nonlinear—see Figure 3 [3,38]. It can be noticed that, for the considered MR fluids, the value of relative magnetic permeability μ*_r_* is less than 10.

In the proposed approach, the magnetic properties of magnetic soft materials and PMs are expressed by Equations (2) and (3), respectively:(2)B=μH,
(3)$$B=\mu_{0}\left(H+M\right),$$
where μ_0_ is the magnetic permeability of the vacuum, **B** represents the vector of magnetic flux density, **H** is the vector of magnetic field intensity, and **M** represents the magnetization vector.

Since most electrical devices are powered from a voltage source, the electromagnetic field equations must be supplemented with equations describing the current flow in the excitation circuit windings:(4)$$u=Ri+\frac{d}{dt}\Psi ,$$
where **u** is the supply voltage vector, **R** is the matrix of loop resistances, **i** is the vector of loop current, and Ψ is the vector of linkage fluxes. The vector Ψ is determined on the basis of the field model, i.e., by solving Equations (1)–(3).

In order to describe fluid dynamics, a phenomenological approach has been used. In this approach, the fluid is treated as a continuum, the properties of which are described as density ρ, dynamic viscosity ξ, and magnetic permeability μ [4,7,21,30,33], as well as electric conductivity. Since an MRF is a suspension of ferromagnetic particles coated by a nonconductive surface agent in a nonconductive carrier liquid, for the low-frequency magnetic field, the conductivity of an MRF can be neglected [39]. In the proposed model, the laminar flow of a noncompressible fluid with no mass sources was considered [8,40,41,42,43]. Since the gravitational field-induced forces acting on the fluid are negligibly small compared to the forces causing the fluid to move in the brake, they were omitted from consideration. It also is assumed that the flow of the fluid occurs only in the circumferential direction (υ-direction—see Figure 2) and is caused by the rotary motion of the brake rotor with angular speed ω. For such conditions, the motion equation of the fluid can be written as [7,40,42]:(5)$$\frac{\partial }{\partial r}\left(\frac{\xi_{z}}{l}\frac{\partial \phi }{\partial r}\right)+\frac{\partial }{\partial z}\left(\frac{\xi_{z}}{l}\frac{\partial \phi }{\partial z}\right)=\frac{\rho }{l}\frac{\partial \phi }{\partial t},$$
where $\phi =2\pi rv_{\upsilon }$, $v_{\upsilon }$ is the component of velocity v in the υ-direction, $\xi_{z}$ is the equivalent dynamic viscosity of the fluid, and ρ is the fluid density.

In the case of the studied brake, Equation (5) should be solved by taking into account the nonslip boundary conditions $v_{\upsilon }=r\omega$ and $v_{\upsilon }=0$ on the surface of the brake rotor and brake frame, respectively. 

In the elaborated field model for the rheological properties of an MR fluid, the equivalent dynamic viscosity of the fluid has been applied [6,42,43]. In order to calculate $\xi_{z}$ of the fluid, the rheological properties of the MR fluid are taken into account. MR fluids exhibit characteristics of non-Newtonian fluids. The Bingham model can effectively describe the properties of such fluids [20,40,43]. An illustration of the dependence of τ on velocity gradient *D* [6,43] and flux density module *B* in MRF for a one-dimensional (1D) model of the fluid is presented in Figure 4. It can be noticed that the MR fluid behaves like a solid body for shear stress $\left|\tau \right|\leq \tau_{0}\left(B\right)$ and like a body of plastic viscosity ξ_p_ for $\left|\tau \right|>\tau_{0}\left(B\right)$, where ξ_p_ = tg(β) and $\tau_{0}\left(B\right)$ exhibit the yield stress of the fluid exposed to the magnetic flux density *B*.

The impact of the magnetic flux density *B* on fluid yield stress τ_0_ for a few selected types of MRFs is shown in Figure 5 [3,38]. It can be noticed that the yield stress changes in saturation at different values of *B.* It should also be highlighted that the response of the fluid rheological properties to a change in the magnetic field is fast and occurs in microseconds [3,4,9].

In the proposed field model of an MR fluid, the equivalent value of the dynamic viscosity of the fluid can be expressed as [6,43]:(6a)$$\xi_{z}=\xi_{p}+\tau_{0}\left(B\right)/\left\|D\right\|\;\;for\;\;\left\|\tau \right\|>\tau_{0}\left(B\right),$$
(6b)$$\xi_{z}=\infty \;\;for\;\;\left\|\tau \right\|\leq \tau_{0}\left(B\right).$$

The yield stress τ_0_(*B*) in Equation (6a,b) is calculated on the basis of the vector distribution of magnetic flux density **B**, determined by solving an electromagnetic field model, i.e., Equations (1) and (4). The norms $\left\|\tau \right\|,\;\left\|D\right\|$ of the stress tensor **τ** and the share rate tensor **D**, respectively, refs. [8,42] may be written as:(7a)$$\left\|\tau \right\|={\left(\frac{1}{2}\sum_{i=1}^{2}\sum_{j=1}^{2}\tau_{i,j}^{2}\right)}^{1/2},$$
(7b)$$\left\|D\right\|={\left(\frac{1}{2}\sum_{i=1}^{2}\sum_{j=1}^{2}D_{i,j}^{2}\right)}^{1/2},$$
where
(8a)$$D_{i,j}=0\;\;for\;\;\left\|\tau \right\|\leq \tau_{0}\left(B\right),$$
(8b)$$D=0.5\;[\nabla v+{\left(\nabla v\right)}^{T}]\;\;\;for\;\;\left\|\tau \right\|>\tau_{0}(B),$$
(8c)$$\tau_{i,j}=\left(\xi_{p}+\tau_{0}\left(B\right)/\left\|D\right\|\right)\;D_{i,j}\;\;for\;\;\left\|\tau \right\|>\tau_{0}\left(B\right).$$

A typical family of characteristics $\xi_{z}\left(\tau ,B\right)$ for a 1D model of the fluid is shown in Figure 6.

As mentioned above, the conductivity γ of ferromagnetic elements of the magnetic circuit and the magnetic permeability of material μ as well as the resistances of the windings and the rheological properties of the fluid depend on the temperature ϑ. In order to take into account the discussed dependencies, the coupled field model of phenomena in the studied MR device should be supplemented with the equation describing the temperature distribution. Since the Taylor number *T_a_* for the considered brake design is much smaller than the critical *T_ac_,* there are no conditions for the convective effect in the MRF. For this reason, the heat transfer problem can be simplified to temperature diffusion and can be expressed by the following equation assuming cylindrical symmetry [24,25,28]:(9)$$\frac{1}{r}\frac{\partial }{\partial r}\left(kr\frac{\partial \vartheta }{\partial r}\right)+\frac{\partial }{\partial z}\left(k\frac{\partial \vartheta }{\partial z}\right)=\rho \;c\frac{\partial \vartheta }{\partial t}-\;p_{h},$$
where $k=k\left(\vartheta \right)$ represents the thermal conductivity and heat capacity is described by parameter *c*.

The symbol *p_h_* describes the heat sources resulting from the dissipation of (a) mechanical energy in the viscous fluid, (b) electrical energy in the winding and conductive elements of the transducer, and (c) mechanical energy in the seals and bearings. The heat sources in the studied HEMRFB can be calculated from the following formula [28,33]:(10)$$p_{h}=\left\{\begin{array}{c} \frac{\omega T_{b}}{V_{b}}\;\;\;\;\;\;\;\;\;\;\;\;\;\;\;\;\;\;\;\;\;\;\;\;\;\;\;\;\;\;\;\;\;\;\;\;\;\;\;\;\;\;\;\;\;\;\;for\;the\;bearing\;\\ \frac{\omega T_{s}}{V_{s}}\;\;\;\;\;\;\;\;\;\;\;\;\;\;\;\;\;\;\;\;\;\;\;\;\;\;\;\;\;\;\;\;\;\;\;\;\;\;\;\;\;\;\;\;\;\;\;\;\;\;\;\;\;for\;the\;seal\\ \xi_{z}\left\{{\left[r\frac{\partial }{\partial r}\left(\frac{v_{\upsilon }}{r}\right)\right]}^{2}+{\left(\frac{\partial v_{\upsilon }}{\partial z}\right)}^{2}\right\}\;\;\;\;\;\;\;\;\;\;\;\;\;\;\;\;\;\;\;\;\;\;\;\;\;for\;the\;fluid\\ \frac{J^{2}}{\gamma }\;for\;\mathrm{the}\;\mathrm{windings}\;\;and\;\;\mathrm{the}\;\;\mathrm{region}\;\;\mathrm{with}\;\;\mathrm{eddy}\;\mathrm{currents} \end{array}\;\;\;\right.,$$
where $T_{s}$, $V_{s}$, $T_{b}$, $V_{b}$, and *J* are the friction torques, volumes of the seal and bearing, and current density.

Problem (9) must be solved by taking into account a natural boundary condition on the external surface of the considered MR brake [7,42]. In the proposed approach, we assume that heat flux *q_n_* at the boundary is proportional to the difference in the temperatures $\left(\vartheta -\vartheta_{a}\right)$ between the external surface of the transducer and the environment: $q_{n}=k\partial \vartheta /\partial n=-h\left(\vartheta -\vartheta_{a}\right).$ In this equation, *h* represents the coefficient of the convection heat transfer, and ∂/∂n represents the derivative in the direction normal to the boundary surface.

In the performance analysis of the proposed HEMRFB, Equations (1), (4), (5) and (9) describing the electromagnetic fluid flow and thermal phenomena have to be solved together with the equation of brake dynamics. For the considered HEMRFB, the equation for the dynamics of movable elements can be expressed as follows: (11)$$J_{b}\frac{d\omega }{dt}+T_{c}+T_{fr}=T_{in},$$
where *J_b_* represents the inertia of the brake rotor and the load, $T_{c}\left(v_{\upsilon },B\right)$ is the braking torque, which is determined from the field model of MR fluid flow, and *T_fr_* represents the total braking torque generated by the brake bearings and seals, while *T_in_* represents the driving torque.

The value of torque *T_c_* was calculated from the integral of the stress tensors along the closed surface s placed in the MR fluid and covering the rotor, using the formula:(12)$$T_{c}=\underset{s}{∯}r\left(\tau_{\upsilon }+\tau_{e\upsilon }\right)ds,$$
where the vectors **τ***_e_*_υ,_ **τ**_υ_ in the above formula describe, respectively, tangentially to the external surface **s** of the rotor, the stress in the fluid and the electromagnetic stress acting in the direction υ.

## 3. FEM Formulation

The equations of the presented field model of the MR brake with hybrid excitation are coupled through the viscosity function $\xi_{z}=\xi_{z}(B,v_{\upsilon })$, torque $T_{c}\left(B,v_{\upsilon }\right)$, and dependence of material properties on temperature, as well as through the boundary condition $v_{\upsilon }=r\omega$ on the surface of the moving parts. Therefore, the obtained equations should be solved simultaneously. In order to calculate nonlinear Equations (1), (4), (5) and (9) of the coupled phenomena model, only approximate methods can be applied based on the discretization of space and time [35,36,37]. The finite element method in connection with a Galerkin approach allows for obtaining the following system of matrix equations, which describes the distribution of electromagnetic fields, currents in windings, velocity fields in an MR fluid, and temperature distribution:(13)$$\left[\begin{array}{cc} S+Gp & -w\\ -w^{T}p & -R \end{array}\right]\left[\begin{array}{c} \varphi \\ i \end{array}\right]=\left[\begin{array}{c} M_{m}\\ -u \end{array}\right],$$
(14)$$[S_{}^{\prime }+G^{\prime }p]\;\phi =H,$$
(15)$$\left(S^{\prime \prime }+K+G^{\prime \prime }p\right)\;\Theta =P+F,$$
where **S**, **S**′, and **S**″ are the coefficient matrices for the equations describing the magnetic, hydrodynamic, and thermodynamic field, respectively; **φ**, **ϕ**, and **θ** are the vectors of the nodal values φ, ϕ, and ϑ, respectively, *p* = *d*/*d*t, **G** is the matrix of conductances of elementary rings formed by the mesh, ***M****_m_* is the vector of magnetomotive force in the regions with permanent magnets, **w***^T^* is the matrix that transforms the potentials φ into flux linkages in the windings, **G**′ is the matrix whose elements depend on the dimensions of the elementary rings and fluid density ρ, **G**″ is the matrix whose elements depend on the dimensions of the elementary rings and heat capacity *c*, **P** is the vector of the nodal heat sources, **H** is the vector of boundary conditions, and **K** and **F** represent the coefficient matrices, which describe the heat transport to the surroundings of the brake [25,28,42].

In the formulated problem, the elements of matrix **R*,*** i.e., the matrix describing the resistances of windings, the elements of conductance matrix **G*,*** and the elements of matrix **S** in Equation (13) depend on temperature. Calculating the values of elements of matrices **R** and **G**, it is assumed that the winding resistance *R*(ϑ) and the conductivity γ(ϑ) of steel are obtained from the well-known relations
(16)$$R\left(\vartheta \right)=R_{20^{{}^{\circ}}C}\left[1+\alpha_{R}\left(\vartheta -20\right)\right],$$
(17)$$\gamma \left(\vartheta \right)=\frac{\gamma_{20^{{}^{\circ}}C}}{1+\alpha_{\gamma }\left(\vartheta -20\right)},$$
where ϑ is the temperature on the Celsius scale, *R*_20°C_ is the resistance of the winding at 20 °C, γ_20°C_ is the conductivity of steel at 20 °C, and α*_R_* and α_γ_ are the thermal coefficients of the resistivity of copper and steel, respectively [7,25,28].

The matrix **S** can be treated as a “magnetic” stiffness matrix of elements that depend on the geometry of the applied mesh and magnetic permeability μ of materials. In the proposed approach, it is taken into account that the magnetic permeability of the applied materials is a function of the two physical quantities: material temperature ϑ and magnetic field intensity *H*. The value of μ of the material can be determined based on the family of *B*(*H*) curves measured at a given temperature ϑ. To calculate μ, these characteristics can be given in a tabular form. However, to minimize the calculation time, it is beneficial to introduce the analytical formulas of μ (*H,* ϑ) that approximate the measured characteristics. The formulas approximating the influence of temperature and magnetic field intensity *H* on permeability should reproduce the phenomenon of losing the ferromagnetic properties of the material after reaching the Curie temperature ϑ*_C_*. In such a case, the relative permeability should be equal to unity. The following expression was implemented in the proposed approach for calculating the magnetic permeability of the ferromagnetic materials [25,44]:(18)$$\mu \left(H,\vartheta \right)=\mu_{0}\left(1+\frac{c_{1}}{1+\frac{H}{c_{2}}}\sqrt{\frac{\vartheta_{C}-\vartheta }{\vartheta_{C}}}\right),$$
where the values of parameters *c*_1_ and *c*_2_ were selected in such a way so that the curve relation presented in (18) reflected the measurement results as closely as possible.

The proposed model and developed algorithm assume that temperature influences the value of the thermal conductivity of materials *k*. When determining the elements of matrix **S**″, it is presumed that the abovementioned relation is calculated according to the following formula [25,28]:(19)$$k\left(\vartheta \right)=k_{20^{{}^{\circ}}C}\left[1+\chi \left(\vartheta -20\right)\right],$$
where symbol *k*_20°C_ represents the material thermal conductivity at 20 °C and χ is the thermal conductivity coefficient. 

The most convenient way, which takes into account the influence of temperature on the PM material properties, is to use a set of demagnetization characteristics determined for several temperatures. An example of a set of demagnetization characteristics provided for the applied material VACODYM HR 677 is shown in Figure 7 [45]. 

Temperature also affects the rheological properties of an MR fluid, as confirmed by tests of fluids MRF 132LD and MRF 132AD performed at the Poznan University of Technology. The exemplary results of the test presenting the influence of temperature ϑ and magnetic flux density *B* on shear stress for fluid MRF 132LD are given in Figure 8. It can be seen that the decrease in stress is roughly proportional to the increase in temperature. The influence of temperature ϑ and velocity of deformation *D* on the equivalent viscosity ξ*_z_* is shown in Figure 9. This figure presents that the change in viscosity as a function of temperature, for the constant value of magnetic flux density *B* and velocity of deformation *D*, can be approximated by means of the linear function:(20)$$\xi_{z}\left(\vartheta \right)=\xi_{z}{\left(\vartheta_{o}\right)}_{}^{}\left[1-a_{\xi }\left(\vartheta -\vartheta_{o}\right)\right],$$
where α_ξ_ is the temperature coefficient of viscosity changes and ξ*_z_* (ϑ_0_, *B*) is the viscosity for the reference temperature ϑ_0_ and magnetic flux density *B.*

Equation (20) is used to calculate the elements of matrices **S**′ and **H**, presented in Equation (14). In the elaborated model of coupled phenomena, the influence of temperature on the material heat capacity *c* and the influence of mechanical stress in magnetic materials on magnetic permeability μ are neglected.

In order to solve the equations describing the discussed field model of an MR transducer in the time domain, the authors applied the time-stepping method. The backward difference scheme was introduced for the approximation of the time derivatives in Equations (13)–(15).

The following system of nonlinear algebraic matrix equations is obtained by time discretization:(21)$$\left[\begin{array}{cc} S_{n}+\Delta t^{-1}G_{n} & -w\\ -w^{T} & -\Delta tR_{n} \end{array}\right]\left[\begin{array}{c} \varphi_{n}\\ i_{n} \end{array}\right]=\left[\begin{array}{c} M_{mn}+\Delta t^{-1}G_{n}\varphi_{n-1}\\ -\Delta tu_{n}-w^{T}\varphi_{n-1} \end{array}\right],$$
(22)$$[S_{n}^{\prime }+\Delta t^{-1}G_{n}^{\prime }]\;\phi_{n}=\Delta t^{-1}G_{n}^{\prime }\phi_{n-1},$$
(23)$$[S_{n}^{\prime \prime }+K_{n}+\Delta t^{-1}G_{n}^{\prime \prime }]\;\Theta_{n}=\Delta t^{-1}G_{n}^{\prime \prime }\Theta_{n-1}+P_{n}+F_{n},$$
where subscript *n* denotes the current time step and Δ*t* is the length of the time step.

To solve the dynamics of the movable elements of the brake (11), the time derivatives were approximated using the following explicit formula [33]:(24)$$J_{b}\left(\alpha_{n+1}-2\alpha_{n}+\alpha_{n-1}\right)/{\left(\Delta t\right)}^{2}=T_{in,\;n}-T_{c,\;n}-T_{fr,n},$$
where α is the angular rotor position, while *T_in,n_ = T_in_*(*t_n_*), *T_c,n_ = T_c_*(*t_n_*), and *T_fr,n_ = T_fr_*(*t_n_*).

For the applied numerical scheme, the angular velocity ω of the rotor can be calculated as follows:(25)$$\omega \left(t_{n}+0.5\Delta t\right)=\left(\alpha_{n+1}-\alpha_{n}\right)/\Delta t,$$

The braking torque *T_c,n_* in Equation (24) is calculated using expression (12). In the studied HEMRFB, the axial symmetry of the magnetic field distributions was assumed; therefore, the component *B*_υ_ of the magnetic flux density *B* is equal to zero and the electromagnetic component in (12) τ_eυ_, resulting from the Maxwell stress tensor, is also equal to zero.

The main problem in obtaining a numerical solution to the MR fluid flow field given in Equation (5) is the presence of a surface separating the shear and nonshear fluid regions [18]. The location of this surface is not known in advance, i.e., prior to the velocity field calculations [6]. The use of the equivalent dynamic viscosity formulas described earlier eliminates the need to track the surface separating the two flow regions and simplifies the solution. However, this leads to singularity since the equivalent dynamic viscosity ξ*_z_* reaches an infinite value in the regions where the share rate is zero $\left(\left\|D\right\|=0\right)$, i.e., in the regions where the fluid behaves as a solid. To avoid this problem, Equation (6a,b) are replaced by the formula proposed in [6]:(26)$$\xi_{z}=\xi_{p}+\frac{\tau_{0}}{\left\|D\right\|}\;\left(1-e^{-m\left\|D\right\|}\;\right)\;,$$
where *m* is the exponential growth parameter.

This approach provided a good approximation of the Bingham fluid properties for both low and high shear stresses τ. Extensive numerical experiments led to the determination of *m* = 100 as a value large enough to obtain accurate solutions [6,20]. Due to the coupling of the considered phenomena and the nonlinearity of material properties, Equations (21)–(25) should be solved simultaneously. In the developed algorithm and computer code, these equations are solved using Newton’s iterative method and the block over-relaxation procedure [25,29,46,47].

## 4. Results

The proposed field model of coupled phenomena and the elaborated algorithm for solving the equations of the formulated model were implemented in the form of a developed computer program written in the Object Pascal programming language. The developed tool allows for manual discretization of the considered transducer geometry as well as import of discretization meshes from the GiD environment [48]. To solve the equations of the model, the iterative conjugate gradient method was implemented to solve the systems of equations obtained by FEM formulation. The developed software was applied to analyze the operation states of the HEMRFB shown in Figure 1. To prove model accuracy, a prototype of the hybrid brake and a computer-aided experimental setup were designed and built. The experimental setup components and the brake prototype are given in Figure 10.

The designed brake contained magnetorheological fluids produced by Lord Corporation and marked with the symbol MRF 132AD. The external diameter of the elaborated brake was 250 mm. In the analysis of the MR brake, the influence of temperature on the electromagnetic and rheological material properties was taken into account. Moreover, the same mesh was used in the calculation of the electromagnetic and thermal fields. The paper analyzed the influence of mesh density on the results of the calculations. This density was gradually increased until a difference between two consecutively calculated values of currents or torques was observed. In addition, it was assumed that the delay values for which the fluid reacts to the changes in the magnetic field are neglected. 

First, the elaborated authored program was applied to calculate the electromagnetic field, the velocity field of the fluid, and the braking torque for the current value in the winding *I* = 0 A. The calculations were performed for the rotational speed of the brake shaft equal to 475 rpm. The results of the calculations are given in Figure 11a. The value of antitorque was calculated for the operating state, which was equal to 120 Nm. Next, the value of the current at which the brake is turned off was determined. This operation state was obtained for the value of the excitation current *I* equal to 2.3 A. For this value of the current, the braking torque generated by the brake was equal to 2.8 Nm. In Figure 11b, one can observe the distribution of magnetic and velocity fields in the case when the brake is switched off.

In order to determine the steady-state control characteristic *T*(*I*) of the considered brake, the relationship between the braking torque *T* and the exciting current *I* at a constant rotational speed was tested. The measurements were repeated for different values of rotational speed in both rotational directions of the rotor. It has been found that the direction of rotation does not have any influence on the torque value. When calculating the *T*(*I*) *= T_fr_ + T_c_*(*I*) characteristic, it was assumed that the component *T_c_*(*I*) is determined on the basis of the stress distribution in the MR fluid—see Equation (12). The component *T_fr_* was obtained based on the measurements performed at the test stand. When determining the total braking torque *T_fr_* produced by seals and bearings, the brake was not filled with an MR fluid. The comparison of the calculated and measured control characteristics for rated rotational speed is shown in Figure 12. The change in the current flow direction results in an increase in antitorque with respect to the antitorque obtained for *I* = 0 A. Good concordance between the calculations and measurements has been obtained.

Next, the transient states of the hybrid MR brake were investigated. The reaction time, determined during the process of switching off the supply voltage, was analyzed. It was assumed that before switching off the voltage, the current in the coil was equal to 2.3 A. The measured and calculated waveforms of the current and torque are shown in Figure 13. The distributions of the magnetic field, the velocity field, and the isothermal lines for selected time instants, obtained for the considered operating state, are presented in Figure 14.

A heating test was carried out to define the effect of temperature on the brake and its parameters during a typical work cycle. In the investigations, a typical duty cycle of the elevator was mapped as 180 runs per hour with a 40% share of traveling time in the total cycle time (Figure 15). The cycle time was set to 20 s. The time characteristic of the rotational speed of the brake shaft during the heating process is presented in Figure 15. The temperature of the brake was measured by means of two semiconductor temperature sensors (Figure 10). The first sensor was placed on the external housing, with the second in the internal aluminum housing. The internal housing was in contact with the MR fluid. Therefore, the temperature measured by the second sensor was close to the fluid temperature. During the heating test, the temperature of the winding was calculated based on the measurement of winding resistance. It was assumed that the maximum value of rotational speed was equal to 400 rpm. The heating test lasted 4.5 h.

The temperature has an influence on the brake parameters. It has been observed that the braking torque in the “warm” state (110 Nm) is about 6% lower than the braking torque in the “cold” state (117 Nm). The decrease in the braking torque along with the increase in temperature is caused by (a) a decrease in fluid viscosity and (b) a decrease in the magnetic flux, generated by permanent magnets. The resistance of the winding in the ambient temperature was equal to 24 Ω. An increase in temperature resulted in a resistance increase of about 15%. The recorded and calculated curves of temperature increase in (a) the housing, (b) MR fluid, and (c) windings are shown in Figure 16. The measurements and calculations were performed at an ambient temperature equal to ϑ*_a_* = 23 °C. The obtained temperature rises Δϑ = ϑ − ϑ*_a_* of the brake subassemblies were less than 23 °C. The obtained increases in temperature are not dangerous to the fluid and permanent magnets. They can be dangerous if the ambient temperature increases.

## 5. Discussion and Conclusions

The paper presents and discusses a coupled field model of phenomena in a magnetorheological brake with hybrid excitation. An algorithm for solving model equations has been proposed. On the basis of the given authored algorithm, computer software has been developed and implemented. The carried-out analysis of the operation of the studied MR brake has considered the nonlinear, rheological, magnetic, and thermal properties of the used materials, eddy currents induced in massive conductive elements, fluid dynamics, and mechanical equilibrium equations. The influence of temperature on the electromagnetic and rheological properties of the used materials has been taken into account. The conducted comparative analysis between the simulation results and measurements of the brake performance shows good concordance and confirms the validity of the model.

The presented field model and the elaborated software enable a more detailed analysis of the phenomena occurring in a brake with an MR fluid than in classical analytical models. According to the authors, the approach proposed in this paper can be helpful in the design and/or optimization process of other types of electromagnetic transducers in which the working medium is an MR fluid (MRF transducers), such as clutches or dampers. Nevertheless, it should be noted that due to the relatively high computational complexity of coupled field analysis (calculation time of a single time step for the studied brake case with a mesh of about 190,000 triangular elements with an Intel I7 processor PC is between 70 to 140 s depending on the number of nonlinear iterations), application of simpler lumped parameter models in the initial stages’ design and optimization calculations of MRF transducers may be convenient. The effectiveness of similar strategies for the design of electromagnetic transducers has been discussed among others in [49], where a tradeoff between computational complexity and model accuracy was discussed in the case of 3D and 2D models of electromagnetic phenomena.

The experimental and simulation results obtained confirm the suitability of the proposed brake for use in elevators (this is confirmed by the tests conducted on its functional and thermal parameters). However, it should be stated that the development of this technology also depends on the “durability” of an MRF. Due to high-reliability requirements, the insufficient durability of rheological parameters may limit the implementation of this technology with time. Another important aspect that needs to be studied before implementation is the analysis of the robustness of the proposed HEMRFB.

## 6. Patents

The design solutions for the research results presented in the paper have been described in a patent Piech, Z.; Szelag, W. Elevator Brake with Magneto-Rheological Fluid. US8631917B2, 21 January 2014.

## Figures and Tables

**Figure 1 sensors-23-00358-f001:**
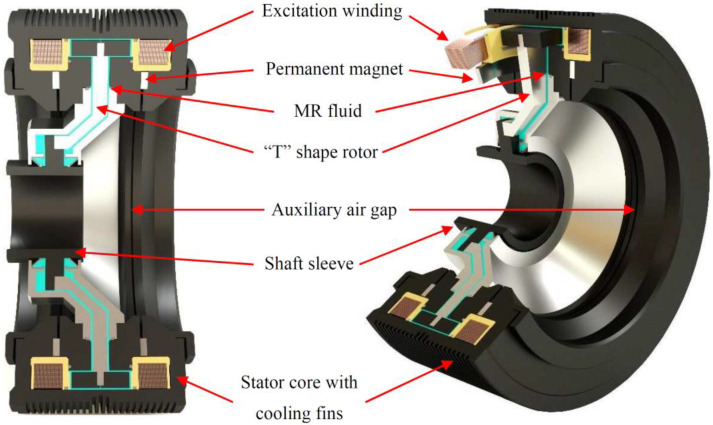
Structure of the proposed hybrid excited brake with magnetorheological fluid.

**Figure 2 sensors-23-00358-f002:**
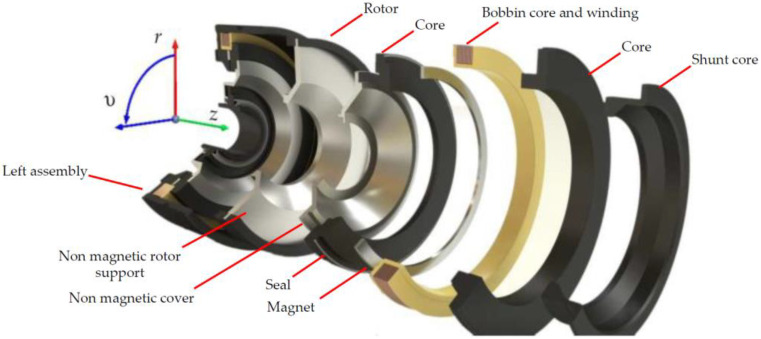
Illustration of cylindrical coordinate system applied in studies of HEMRFB.

**Figure 3 sensors-23-00358-f003:**
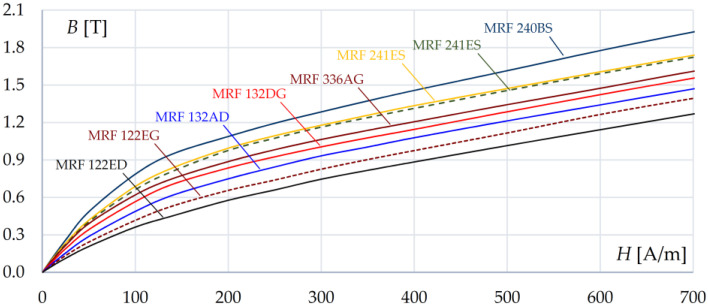
The characteristics *B*(*H*) of selected types of magnetorheological fluids at 20 °C.

**Figure 4 sensors-23-00358-f004:**
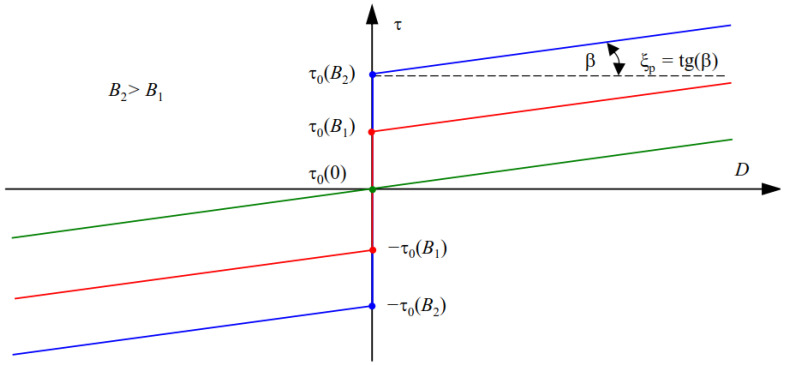
Illustration of dependence of the shear stress in MRF on velocity gradient and magnetic flux density in MRF.

**Figure 5 sensors-23-00358-f005:**
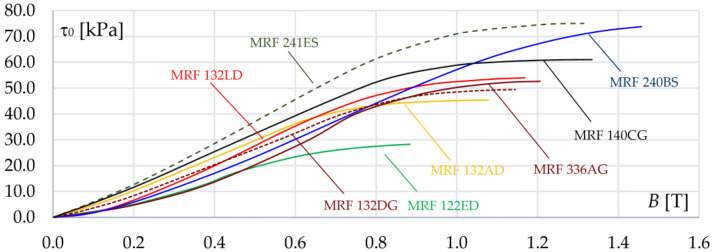
The characteristics τ_0_(*B*) of different magnetorheological fluids at 20 °C.

**Figure 6 sensors-23-00358-f006:**
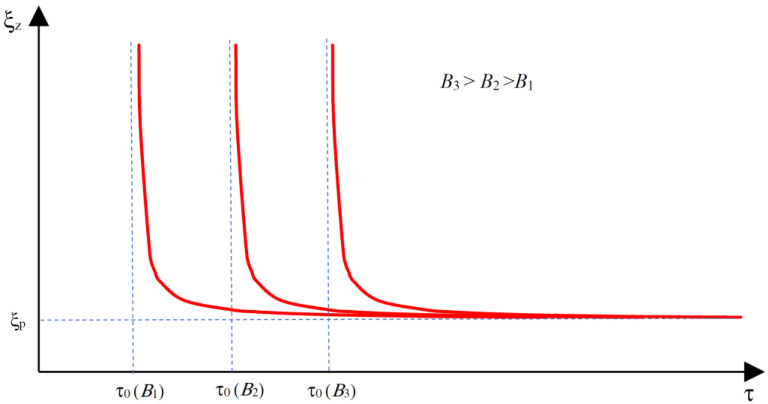
Relationship between ξ*_z_* and stress τ in MR fluid for given value of flux density.

**Figure 7 sensors-23-00358-f007:**
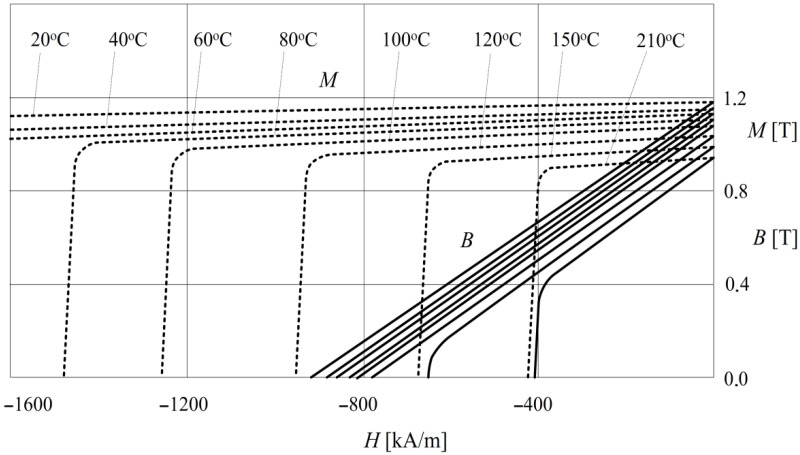
The set of demagnetization characteristics of VACODYM HR 677.

**Figure 8 sensors-23-00358-f008:**
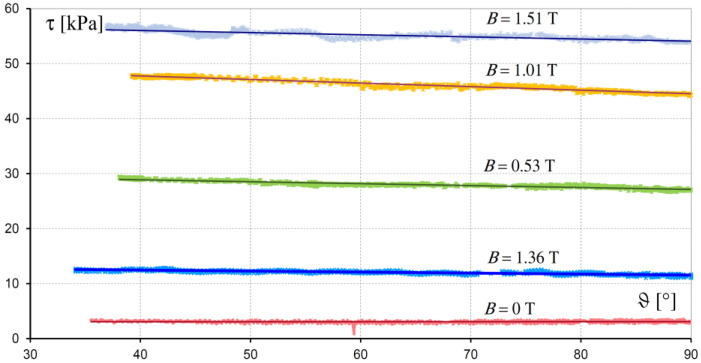
Family of τ(ϑ,B) characteristics for fluid MRF 132LD at *D* = 800 s^−1^.

**Figure 9 sensors-23-00358-f009:**
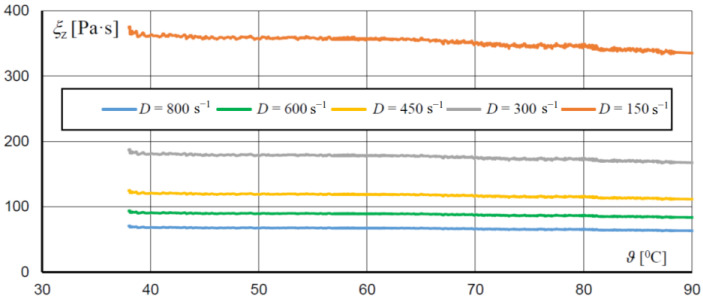
Family of $\xi_{z}(\vartheta ,D)$ characteristics for MRF 132LD fluid at *B* = 1.51 T.

**Figure 10 sensors-23-00358-f010:**
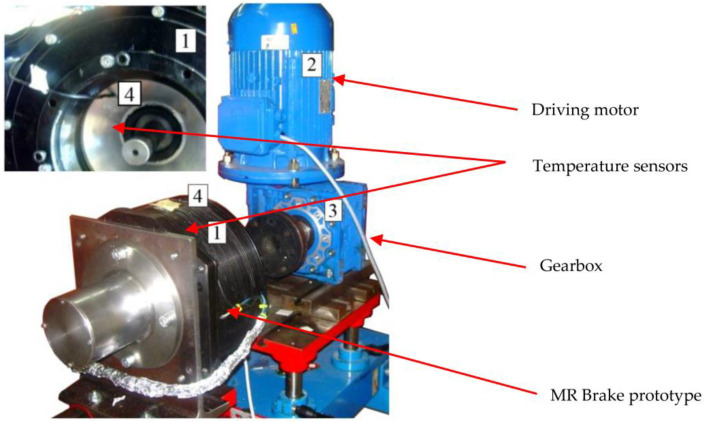
View of the brake test stand; 1—MR brake prototype, 2—Driving motor; 3—Gearbox, 4—Temperature sensors.

**Figure 11 sensors-23-00358-f011:**
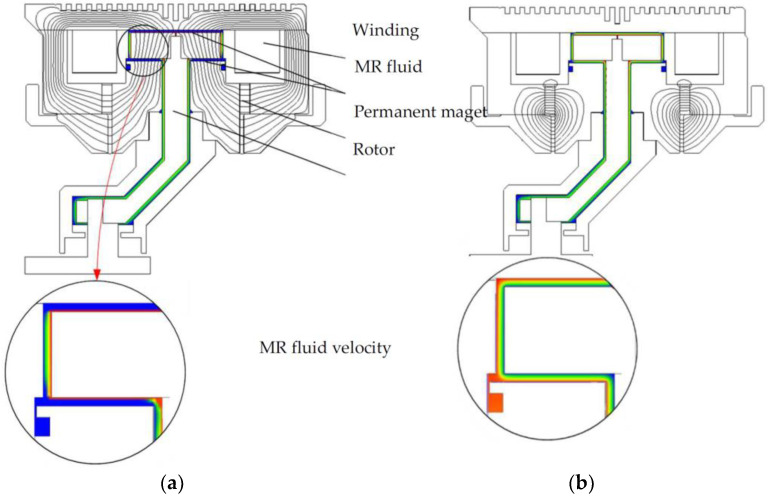
View of magnetic field line distribution in the cross-section of the brake and velocity field in the gap with magnetorheological fluid for current value: (**a**) *I* = 0 and (**b**) *I* = 2.3 A; *n* = 475 rpm.

**Figure 12 sensors-23-00358-f012:**
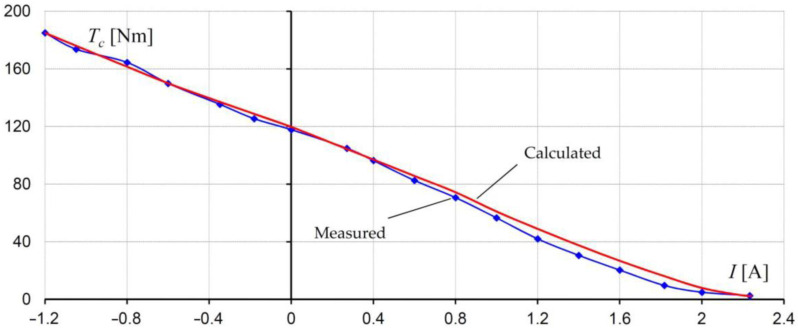
Characteristic of torque as a function of current (control characteristic) of the brake for the rated speed of the rotor *n* = 475 rpm.

**Figure 13 sensors-23-00358-f013:**
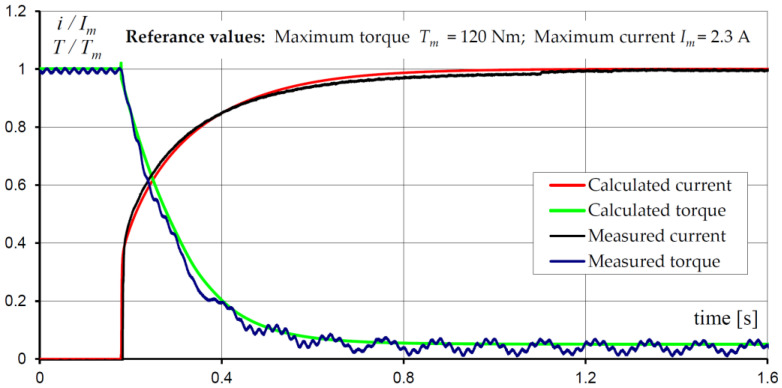
Calculated and measured braking torque *T*/*T_m_* and current *i*/*I_m_* waveforms during switching off the supply voltage for *n* = 100 rpm.

**Figure 14 sensors-23-00358-f014:**
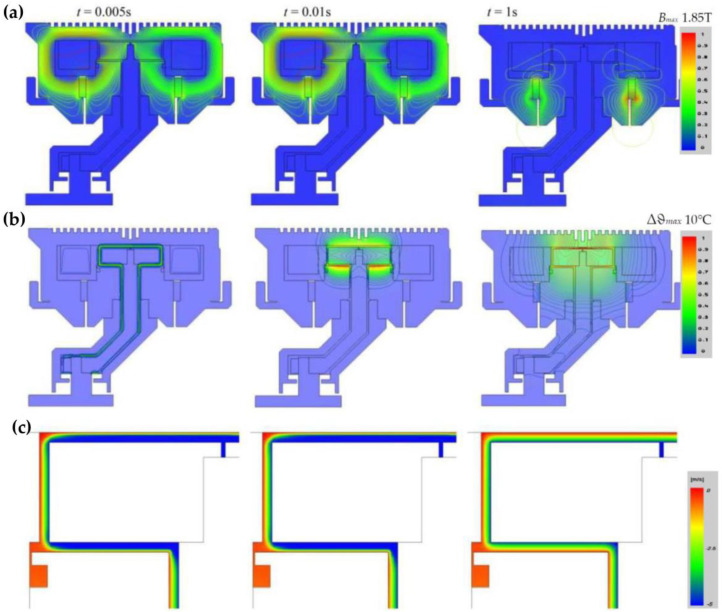
Distributions of (**a**) magnetic field, (**b**) isothermal lines, and (**c**) velocity field in MR fluid in cross-section of the brake for *t* = 0.0001 s, *t* = 0.01 s, and *t* = 1 s (from left to right, respectively).

**Figure 15 sensors-23-00358-f015:**
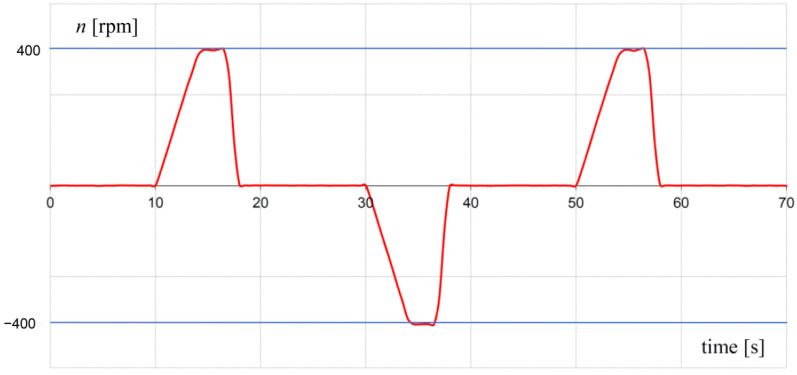
Duty cycle realized during heating test.

**Figure 16 sensors-23-00358-f016:**
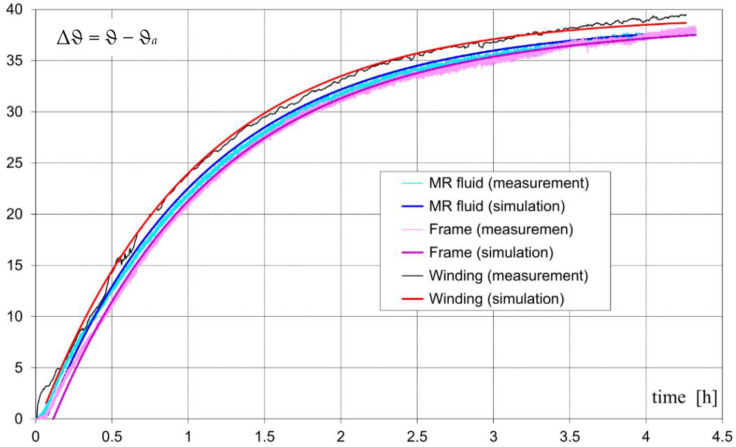
Calculated and measured average rises in temperature of MR fluid brake subassemblies.

## Data Availability

Not applicable.

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
