# Peer review of "Coupled Field Analysis of Phenomena in Hybrid Excited Magnetorheological Fluid Brake"

_sensors, 2022, doi:10.3390/s23010358_

Round 1

Reviewer 1 Report

This paper presents a comprehensive coupled analysis of a hybrid excited magnetorheological fluid brake. This paper is well written, but, in my opinion, a major revision is needed to be considered suitable for publication. My main concerns are the following ones: 

1.- This paper has a lack of originality, because this proposal has been repeatedly presented in previous publications both in journal an in conferences publications, by the same Authors (some of them not cited in this work). In fact, most of the contents of sections 3 and 4 can be found in previous publications, and could be replaced by citing them. The Authors should make clear what are the novelties and advances of this work compared with the state of art.

2.- No information is given about the times and comuting requirements of the simulations presented in this paper. In fact, in previous papers this point was considered to be a problem of the proposed approach.

3.-In equation 9, only the diffusion effect has been considered. What about the effect of the velocity in the thermal distribution?

4.- The references section should include more relevant works on this field, especially those from the Authors.

5.- English language should be thoroughly checked, there are many typos.

Author Response

Dear Reviewer

We would like to thank very much for the valuable and constructive hints and comments helping in improving our paper. We have done our best to improve our manuscript accordingly. Each comment has been carefully considered point by point and responded. We are also grateful for the minor corrections incorporated by the reviewers into the text; we have considered all these suggestions. The answers to the reviewer are listed below. English language and style has been checked and proofread by a skilled in technical language native speaker.

Changes in the text related to the improvement of the article have been highlighted in yellow.

Reviewer 2 Report

This is a good paper that clearly demonstrated the concepts of Magnetorheological Fluid Braking systems strongly supported by equation. I recommend its publication in Open Access in MDPI after a few grammatical edits.

Author Response

(The authors gave the same response as above.)

Reviewer 3 Report

The comments and recommendations see enclosed file

Regards

Author Response

(The authors gave the same response as above.)

Reviewer 4 Report

This work studies the coupling behavior in a hybrid excepted magnetorheological brake via coupled field analysis. Different coupling behaviors are taken into account. The authors used finite element analysis and developed an algorithm as part of the field model of coupled phenomena, for studying the axisymmetric MR brake with hybrid excitation system. A prototype of the brake has also been designed and tested.

The overall paper is written in a good form and I recommend publishing it after the author improves the writing.  

I suggest the authors proofread the paper and have English native speakers help go through the paper. The coherence between sentences can be improved. In the current form, the sentences are written in a similar structure (for example, in both abstract and conclusion), making it monotone and hard to read. Other grammar mistakes need to be avoided, such as the one in line 54, a comma needs to follow "However". Many sentences are very long, e.g., line 82 and line 99, consider adding commas or breaking long sentences into shorter ones. In addition, line 109 and line 113 both starts with "In this paper", which is repetitive. Consider rephrasing or organize the two paragraphs differently.

Additional comments:

1. Figure 2 has many components. Please label what these parts are, otherwise, it does not make too much sense to include it in the main manuscript. 

2. Line 127 claims the magnetization characteristics are nonlinear - see Fig.2. How does Figure 2 suggest the nonlinear characteristics? Please elaborate why nonlinear here. 

3. Label explicitly what parameters the letters stands for in Figure 4.

4. Figure 14 needs scale bar(s)

5. The conclusion paragraph can be rewritten to have better flow between the sentences. Also, it should start with "This/The paper".

Author Response

(The authors gave the same response as above.)

Round 2

Reviewer 1 Report

The Authors havee addressed adequately my concerns, and I think that this paper is suitable for publication in present form.